# A Comprehensive Overview of the Temperature-Dependent Modeling of the High-Power GaN HEMT Technology Using mm-Wave Scattering Parameter Measurements

**Giovanni Crupi** [1,*] **, Mariangela Latino** [2] **, Giovanni Gugliandolo** [2] **, Zlatica Marinković** [3] **, Jialin Cai** [4] **, Gianni Bosi** [5] **, Antonio Raffo** [5] **, Enza Fazio** [6] **and Nicola Donato** [2]

1 Department of Biomedical and Dental Sciences and Morphofunctional Imaging, University of Messina, 98125 Messina, Italy
2 Engineering Department, University of Messina, 98166 Messina, Italy; mariangelacatena.latino@unime.it (M.L.); giovanni.gugliandolo@unime.it (G.G.); ndonato@unime.it (N.D.)
3 Faculty of Electronic Engineering, University of Niš, 18000 Niš, Serbia; zlatica.marinkovic@elfak.ni.ac.rs
4 Key Laboratory of RF Circuit and System, Ministry of Education, College of Electronics and Information, Hangzhou Dianzi University, Hangzhou 310018, China; caijialin@hdu.edu.cn
5 Engineering Department, University of Ferrara, 44122 Ferrara, Italy; gianni.bosi@unife.it (G.B.); rffntn@unife.it (A.R.)
6 Department of Mathematical and Computational Sciences, Physics Science and Earth Sciences, University of Messina, 98166 Messina, Italy; enza.fazio@unime.it
* Correspondence: crupig@unime.it

**Abstract:** The gallium-nitride (GaN) high electron-mobility transistor (HEMT) technology has emerged as an attractive candidate for high-frequency, high-power, and high-temperature applications due to the unique physical characteristics of the GaN material. Over the years, much effort has been spent on measurement-based modeling since accurate models are essential for allowing the use of this advanced transistor technology at its best. The present analysis is focused on the modeling of the scattering (*S*-) parameter measurements for a 0.25 µm GaN HEMT on silicon carbide (SiC) substrate at extreme operating conditions: a large gate width (i.e., the transistor is based on an interdigitated layout consisting of ten fingers, each with a length of 150 µm, resulting in a total gate periphery of 1.5 mm), a high ambient temperature (i.e., from 35 °C up to 200 °C with a step of 55 °C), a high dissipated power (i.e., 5.1 W at 35 °C), and a high frequency in the millimeter-wave range (i.e., from 200 MHz up to 65 GHz with a step of 200 MHz). Three different modeling approaches are investigated: the equivalent-circuit model, artificial neural networks (ANNs), and gated recurrent units (GRUs). As is shown, each modeling approach has its pros and cons that need to be considered, depending on the target performance and their specifications. This implies that an appropriate selection of the transistor modeling approach should be based on discerning and prioritizing the key features that are indeed the most important for a given application.

**Keywords:** artificial neural networks; equivalent circuit; GaN; HEMT; gate recurrent units; high-power; high-temperature; machine learning technique; millimeter-wave frequency; scattering parameter measurements

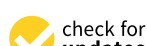

## 1. Introduction

Since their inception in the early 1990s [1], the aluminum-gallium-nitride/gallium-nitride (AlGaN/GaN) high electron-mobility transistor (HEMT) devices have greatly evolved. The GaN-based transistors have been demonstrated to be a very attractive solution for various applications [2–4], such as high-frequency power amplifiers [5–10]; high-power switching systems [11–13]; optoelectronics [14–16]; and, recently, quantum computing electronics [17]. The huge interest in the adoption of this unique and disruptive

III/V semiconductor technology for an ever-growing gamut of applications is due to its extraordinary physical properties [18–22] (e.g., wide bandgap, large breakdown electric field, unprecedented power density, good thermal conductivity, high working temperature, high electron mobility, high electron saturation velocity, strong spontaneous and piezoelectric polarization, and a two-dimensional electron gas (2DEG) with a high sheet carrier density).

As is well known, the extraction of accurate models plays a key role in enabling the use of any advanced transistor technology at its best. Over the years, many efforts have been devoted to the extraction of a small-signal model for the GaN HEMT technology by using scattering (*S*-) parameter measurements since the analysis of the small-signal behavior is considered a crucial prerequisite before investigating both large-signal and noise characteristics. It should be underlined that *S*-parameters can be accurately and straightforwardly measured by using a vector network analyzer (VNA). The equivalent-circuit representation is often adopted [23–35] since it can be used as the cornerstone for the development of both large-signal [36–40] and noise equivalent-circuit models [41–45]. However, as an alternative solution, small-signal behavioral models have been developed to directly mimic the *S*-parameter measurements without the need to determine an equivalent-circuit representation. Typically, artificial neural networks (ANNs) are used as a powerful and effective tool for small-signal behavioral modeling due to their outstanding learning and generalization abilities [46–52]. Another valuable behavioral approach consists of using the standard recurrent neural networks (RNNs) and their advanced variants, such as long-short term memories (LSTM) and gated recurrent units (GRU), which are an extension of the traditional ANNs that became more popular in the recent years due to their superior performance [53–55]. Compared to LSTM, the GRU model has a simpler structure and uses fewer training parameters, thereby resulting in a faster computation time and improved memory usage. Undoubtedly, there are pros and cons to each modeling approach. Therefore, there is no universally best modeling approach but a variety of situations where one or the other approach is better suited, depending on the given constraints and requirements.

With the aim to highlight the advantages and disadvantages of each modeling approach, the present article is focused on developing a comparative analysis. The device under test (DUT) considered in this study is a multi-finger on-wafer GaN HEMT transistor grown on a SiC substrate with a gate length of 0.25 μm and a gate width of 1.5 mm (i.e., $10 \times 150$ μm). The multi-finger layout, also known as interdigitated, is typically used in the realization of transistors for high-power applications. This is because the multi-finger structure, which consists of the parallel connection of multiple gate fingers, allows for enlarging the size of the total gate periphery and, thus, the output current. At the same time, the multi-finger structure allows making the extrinsic gate resistance smaller and bettering the microwave noise performance and the maximum frequency of oscillation ($f_{max}$). Due to the lack of a native GaN substrate, GaN HEMT transistors are commonly realized using various foreign substrates, such as sapphire, silicon (Si), and silicon carbide (SiC). Compared to sapphire and Si, SiC is more expensive, but its excellent thermal conductivity makes it the material of choice for enhancing the power handling capability of GaN HEMTs. An efficient heat spreading is a mandatory requirement, especially when considering a large device operating at high power and high temperature, as in the present study. The *S*-parameters of the DUT were measured with a frequency range spanning from 200 MHz to 65 GHz and at four values of the ambient temperatures ($T_a$), namely 35 °C, 90 °C, 145 °C, and 200 °C. The importance of investigating the impact of the temperature on the behavior of the GaN HEMTs lies in the fact that the device performance can considerably vary with the selected working thermal condition [56–60]. The bias point was fixed at $V_{DS} = 30$ V and $V_{GS} = -3.1$ V, corresponding to a dissipated power of 5.1 W when $T_a$ is set to 35 °C. To model the measured *S*-parameters, three different strategies are considered here: the equivalent-circuit approach, ANNs, and GRUs. The models are compared in terms of accuracy, generalizability, complexity, compactness, and usefulness. The equivalent circuit is a physically meaningful representation, enabling the achievement of deeper insights

into the device physics, and is suitable to be used as a starting point for building both large-signal and noise models, which play a crucial role in a successful design of high power amplifiers (HPAs) and low noise amplifiers (LNAs). ANNs have good learning ability and generalization capabilities, allowing for achieving accurate predictions even for data different from those used in training. The approach based on GRUs allows for accurately fitting the small-signal behavior of the DUT, thus achieving a faithful reproduction of the experiments.

The remainder of this article is structured as follows: Section 2 is aimed at describing the tested GaN-based device and the three considered small-signal modeling strategies; Section 3 is dedicated to presenting and discussing the achieved experimental findings; and the last section summarizes the main conclusions of this comparative measurement-based study.

## 2. DUT and Modelling Methodologies

The DUT is an on-wafer GaN HEMT with a gate length of 0.25 μm (see Figure 1). This transistor is based on an interdigitated layout consisting of ten fingers, each with a length of 150 μm, resulting in a total gate periphery of 1.5 mm. The fabrication process is the GH25-10 technology by United Monolithic Semiconductors (UMS) [61]. The DUT was realized using an AlGaN/GaN heterostructure grown on a SiC substrate with a field plate for power applications. This foundry process, entitling a power density of 4.5 W/mm with a typical $f_T$ of 25 GHz, was optimized for X-band (i.e., 8–12 GHz) high-power applications. The *S*-parameters were measured with a frequency range spanning from 200 MHz to 65 GHz with a 200 MHz step at different $T_a$, namely 35 °C, 90 °C, 145 °C, and 200 °C. In order to move the reference planes from the instrument ports to the probe tips, an off-wafer calibration was performed with line-reflect-match standards on a commercial impedance standard substrate (i.e., a GGB Industries CS-5 ISS). The temperature-dependent experiments were carried out without any heat sink, and a software-controlled thermal chuck was used to set the die temperature. The comparative analysis was performed by considering the bias point given by $V_{DS}$ = 30 V and $V_{GS}$ = −3.1 V. This operating bias condition implies a dissipated power of 5.1 W when considering the ambient temperature of 35 °C.

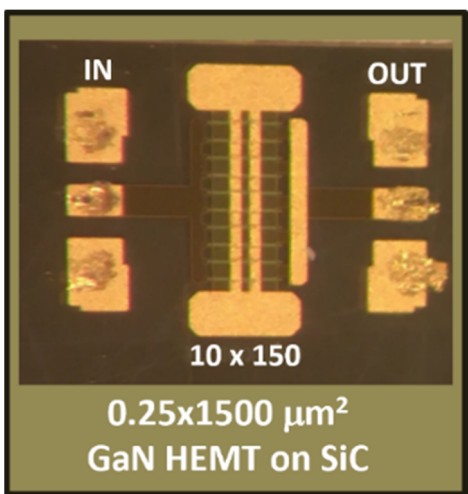

**Figure 1.** Photograph of the tested GaN HEMT on silicon carbide substrate.

It should be underlined that the experiments were performed up to 65 GHz in the V band since the operating frequency of the GaN HEMT technology keeps extending further into the millimeter-wave region, driven by the ever-increasing market demand (e.g., 6G, inter-satellite links) for very high-frequency applications. The measurements were carried out on the GaN HEMT device operating without any heat sink at an elevated ambient temperature of 200 °C, albeit the recommended operating ratings indicate 200 °C as the

peak junction temperature. This allowed for achieving experimental validation of the performance of the studied GaN technology, even beyond the constraints imposed by the device foundry. Furthermore, the Joule heat generated by the high dissipated power leads to a significant rise in the operating channel temperature inside the transistor.

Figure 2 shows the small-signal equivalent-circuit model [32]. This model consists of a physically sound representation since each element of the equivalent circuit is aimed at modeling the electrical characteristics of a particular transistor region. As illustrated in Figure 2, the adopted circuit is composed of eight extrinsic elements, which are assumed to be biased independent and are determined from the "cold" pinch-off $S$-parameters (i.e., $V_{DS}$ = 0 V and $V_{GS}$ = −4 V), and eight intrinsic elements, which are bias dependent and are evaluated from the intrinsic admittance ($Y$-) parameters at the bias point of interest. At such a bias condition, the complexity of the equivalent circuit can be considerably reduced, thereby allowing extraction of the extrinsic equivalent-circuit elements. After applying the de-embedding of the eight extrinsic equivalent-circuit elements by using simple matrix manipulations, the intrinsic $Y$-parameters were calculated from the corresponding $S$-parameters by using the well-known conversion formulas, thereby allowing for the calculation of the intrinsic equivalent-circuit elements. MATLAB software environment is used for implementing the extraction of the equivalent-circuit elements from $S$-parameter measurements.

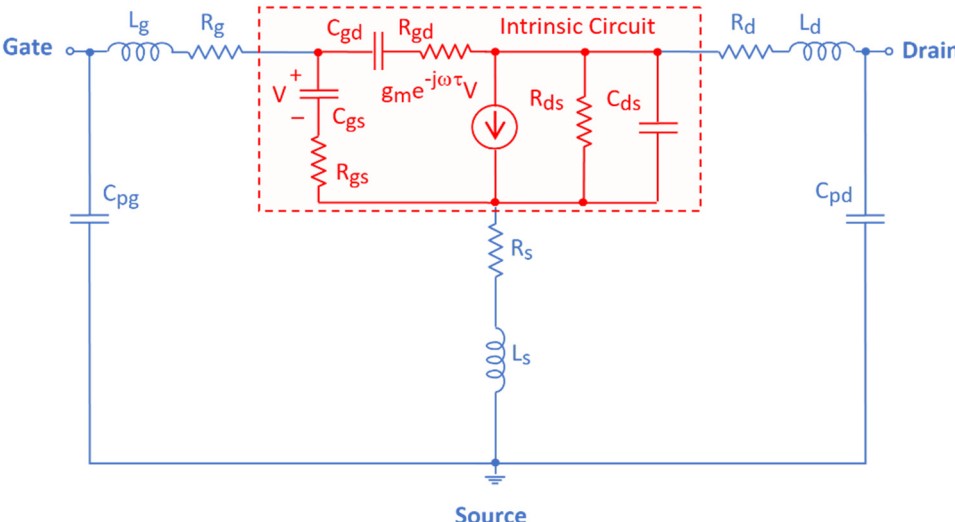

**Figure 2.** Topology of the small-signal equivalent-circuit model for the tested GaN HEMT.

Figure 3 illustrates the ANN temperature-dependent model adopted for modeling the tested GaN-based HEMT [48]. Each $S$-parameter was modeled with a separate neural model. The ANNs are multi-layer perceptron (MLP) networks, which consist of layers of neurons: an input layer, an output layer, and one or more hidden layers. The frequency and the temperature are the two input neurons, whereas the real and imaginary parts of the considered parameter are the two output neurons. The number of hidden layers and their size (i.e., the number of neurons in each hidden layer) were determined during the training. The adopted training algorithm was the Levenberg–Marquardt (L-M) method, which is a modification of the most frequently used optimization backpropagation algorithm [62]. MATLAB software environment was used for training and testing the ANNs. As illustrated in Table 1, the trained ANNs have two hidden layers, each with a different number of neurons. It should be underlined that two separate one-output ANNs were used for modeling the real and imaginary parts of $S_{12}$ because of its more complex behavior that was not possible to reproduce accurately by using a single ANN for both real and imaginary parts. The training data are all the available measurements, with the exception of those at 145 °C, which are used for assessing the model generalization ability.

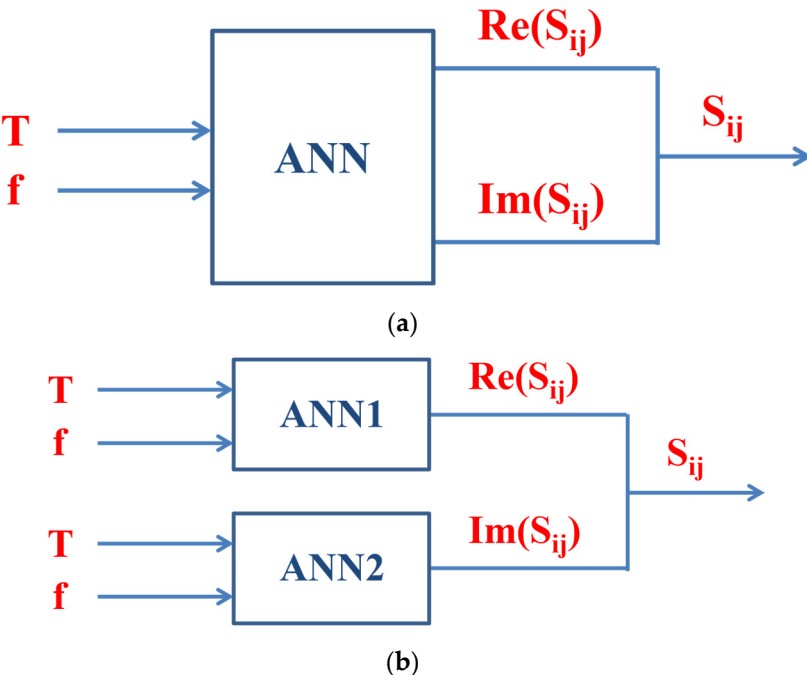

**Figure 3.** Illustration of the model based on using artificial neural networks for the tested GaN HEMT: (**a**) $S_{11}$, $S_{21}$, $S_{22}$, and (**b**) $S_{12}$.

**Table 1.** Structure of the trained artificial neural networks: the number of neurons for the input, first hidden, second hidden, and output layers.

| Parameter | ANN Structure |
|---|---|
| $S_{11}$ | 2-4-4-2 |
| $S_{21}$ | 2-4-4-2 |
| Re ($S_{12}$) | 2-5-5-1 |
| Im ($S_{12}$) | 2-5-4-1 |
| $S_{22}$ | 2-5-5-2 |

Figure 4 illustrates the GRU-based temperature-dependent model adopted for modeling the tested GaN-based HEMT. GRU is served as an efficient method for modeling as its exceeding capability of solving issues such as vanishing and exploding gradients. Compared with techniques such as long short-term memory (LSTM) networks, GRU produces fewer parameters in the model training process, thus making the modeling process more efficient. The proposed GRU model is built and trained using Tensorflow, which is an open-source software library designed for developing and deploying state-of-the-art machine learning algorithms. In the test, each *S*-parameter is modeled with a separate GRU model. As shown in Figure 4, each GRU model has two GRU building blocks, and each GRU blocks share the same structure, with one GRU layer and one dense layer. Figure 5 depicts the internal architecture of a basic unit of the GRU networks. A GRU has two layers with sigmoid functions, corresponding to the so-called reset gate and update gate and a tan-sigmoid layer. Each GRU model consists of a GRU layer and a dense layer. Table 2 shows the detailed layer size information for both the GRU layer and the dense layer for the GRU model. After optimizing weight and bias terms inside the model, two trained models, each for the real part and imaginary part of measured data, are combined to fit and predict the small-signal behavior of the DUT.

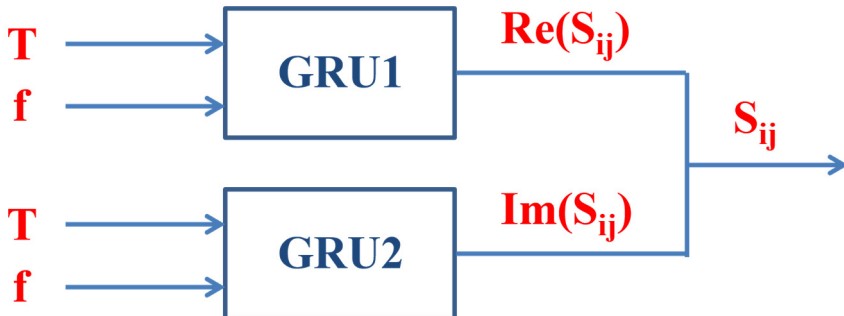

**Figure 4.** Illustration of the model based on using GRU for the tested GaN HEMT.

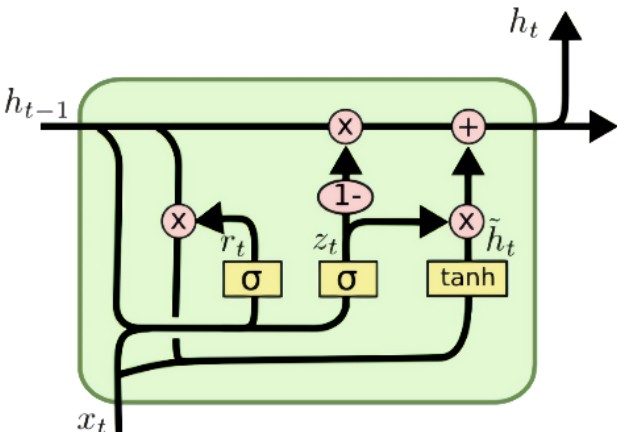

**Figure 5.** Illustration of the internal architecture of a GRU unit. The symbols "$\oplus$" and "$\otimes$" represent the operation of addition and element-wise multiplication, respectively.

**Table 2.** Structure of the trained GRU: size of the GRU layer and the dense layer.

| Parameter | GRU Structure |
|:---:|:---:|
| Re ($S_{ij}$) | 5-1 |
| Im ($S_{ij}$) | 5-1 |

## 3. Experimental Results and Discussion

Figures 6–17 report the comparison between measured and simulated scattering parameters at the four studied values of the ambient temperature. The simulations are achieved by using the three models: equivalent circuit (see Figures 6–9), ANNs (see Figures 10–13), and GRUs (see Figures 14–17). As can be clearly noticed, the measured output reflection coefficient ($S_{22}$) is affected by the kink effect, which consists of an abrupt change in the behavior of this parameter at a certain frequency [63–67]. The GaN HEMT technology is prone to the kink effect in $S_{22}$ due to the relatively high transconductance ($g_m$) [66]. All three investigated models allow for accurately reproducing the measurements over the full studied frequency range and under the four different ambient temperature conditions.

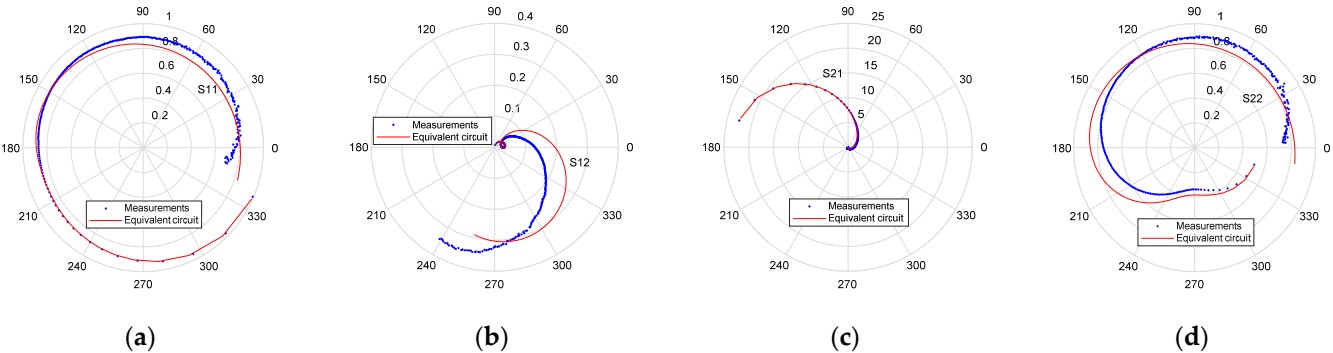

**Figure 6.** Measured (blue symbols) and equivalent-circuit simulated (red lines) behavior of (**a**) $S_{11}$, (**b**) $S_{12}$, (**c**) $S_{21}$, and (**d**) $S_{22}$ with the frequency range spanning from 200 MHz to 65 GHz for the DUT at $V_{DS}$ = 30 V, $V_{GS}$ = −3.1 V, and $T_a$ = 35 °C.

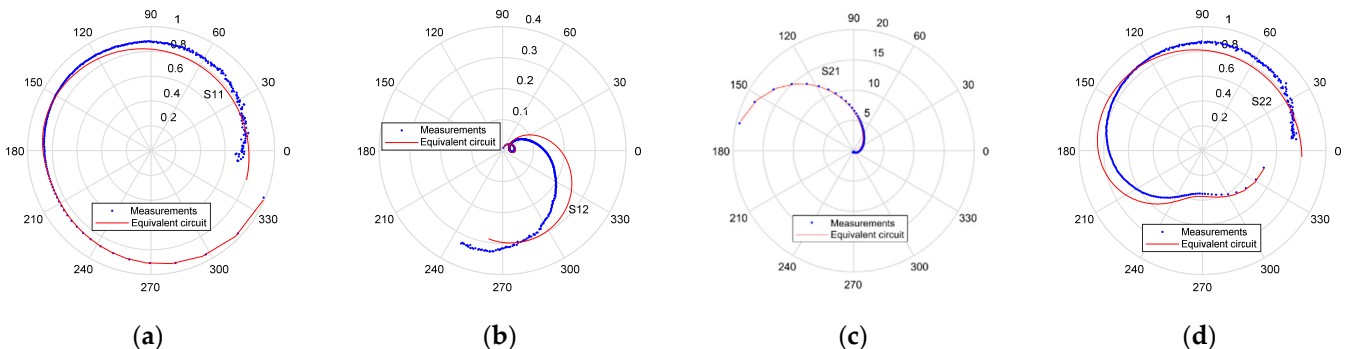

**Figure 7.** Measured (blue symbols) and equivalent-circuit simulated (red lines) behavior of (**a**) $S_{11}$, (**b**) $S_{12}$, (**c**) $S_{21}$, and (**d**) $S_{22}$ with the frequency range spanning from 200 MHz to 65 GHz for the DUT at $V_{DS}$ = 30 V, $V_{GS}$ = −3.1 V, and $T_a$ = 90 °C.

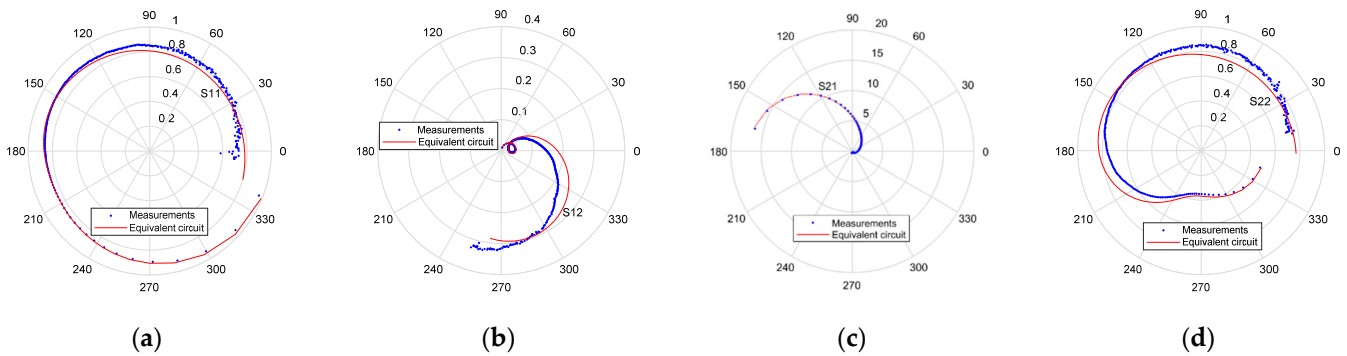

**Figure 8.** Measured (blue symbols) and equivalent-circuit simulated (red lines) behavior of (**a**) $S_{11}$, (**b**) $S_{12}$, (**c**) $S_{21}$, and (**d**) $S_{22}$ with the frequency range spanning from 200 MHz to 65 GHz for the DUT at $V_{DS}$ = 30 V, $V_{GS}$ = −3.1 V, and $T_a$ = 145 °C.

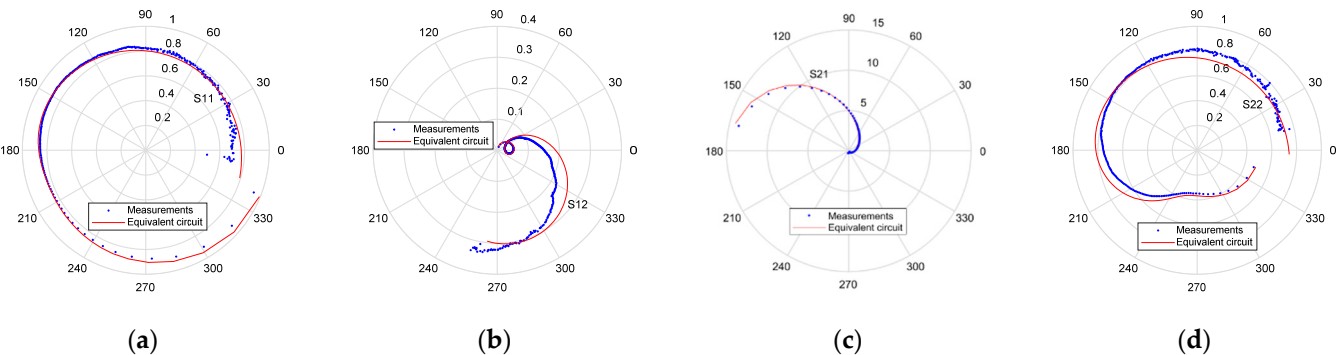

**Figure 9.** Measured (blue symbols) and equivalent-circuit simulated (red lines) behavior of (**a**) $S_{11}$, (**b**) $S_{12}$, (**c**) $S_{21}$, and (**d**) $S_{22}$ with the frequency range spanning from 200 MHz to 65 GHz for the DUT at $V_{DS}$ = 30 V, $V_{GS}$ = −3.1 V, and $T_a$ = 200 °C.

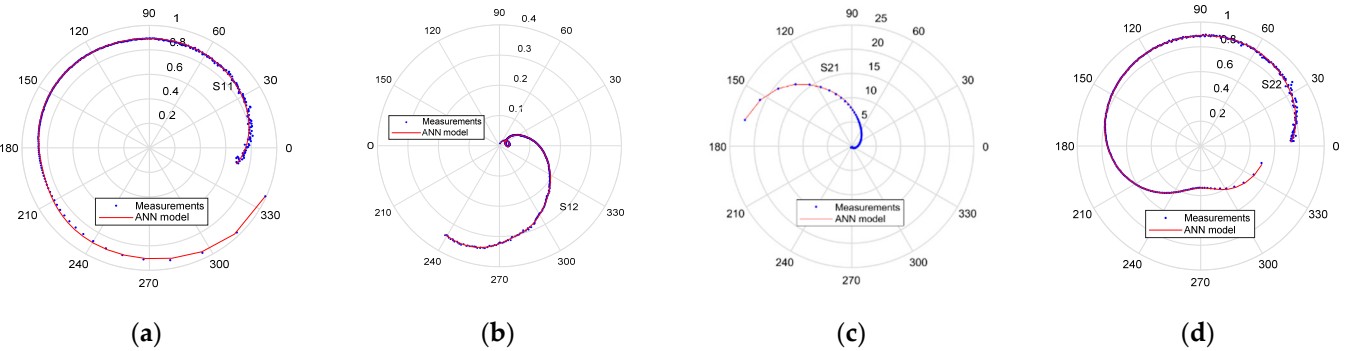

**Figure 10.** Measured (blue symbols) and ANN simulated (red lines) behavior of (**a**) $S_{11}$, (**b**) $S_{12}$, (**c**) $S_{21}$, and (**d**) $S_{22}$ with the frequency range spanning from 200 MHz to 65 GHz for the DUT at $V_{DS}$ = 30 V, $V_{GS}$ = −3.1 V, and $T_a$ = 35 °C.

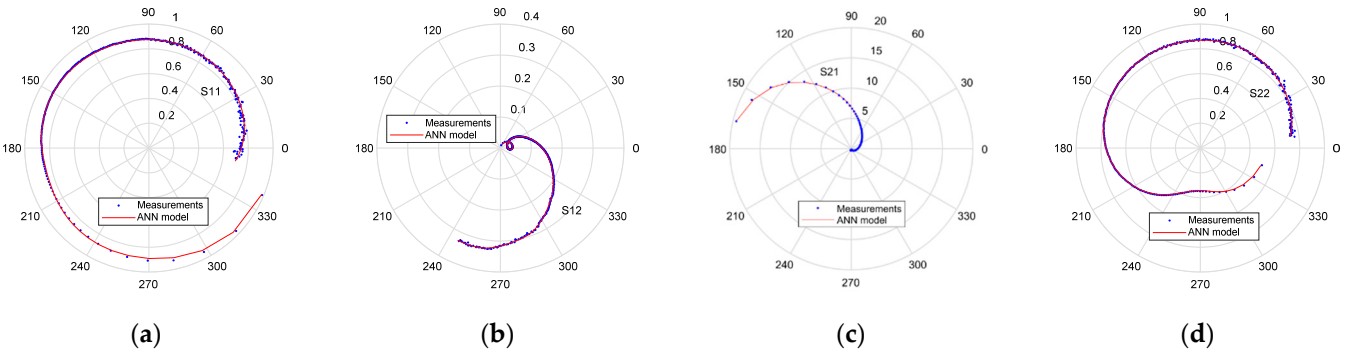

**Figure 11.** Measured (blue symbols) and ANN simulated (red lines) behavior of (**a**) $S_{11}$, (**b**) $S_{12}$, (**c**) $S_{21}$, and (**d**) $S_{22}$ with the frequency range spanning from 200 MHz to 65 GHz for the DUT at $V_{DS}$ = 30 V, $V_{GS}$ = −3.1 V, and $T_a$ = 90 °C.



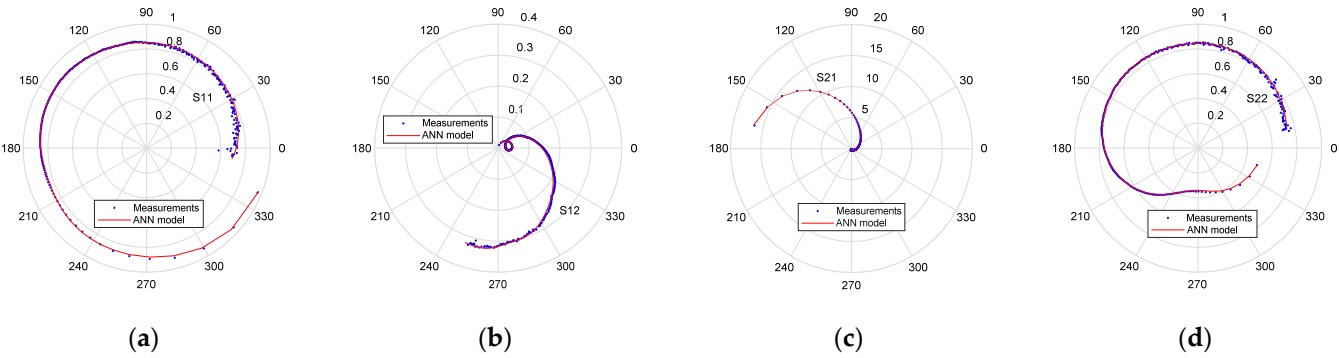

**Figure 12.** Measured (blue symbols) and ANN simulated (red lines) behavior of (**a**) $S_{11}$, (**b**) $S_{12}$, (**c**) $S_{21}$, and (**d**) $S_{22}$ with the frequency range spanning from 200 MHz to 65 GHz for the DUT at $V_{DS}$ = 30 V, $V_{GS}$ = −3.1 V, and $T_a$ = 145 °C.

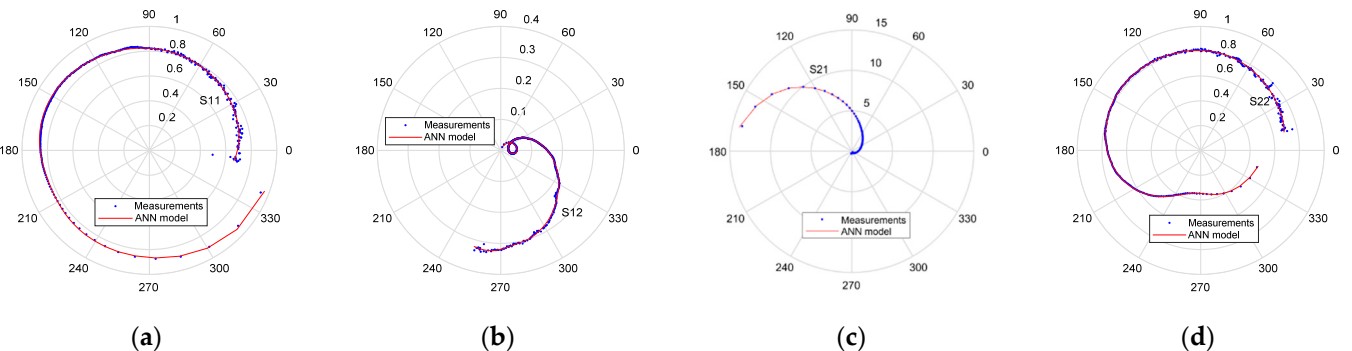

**Figure 13.** Measured (blue symbols) and ANN simulated (red lines) behavior of (**a**) $S_{11}$, (**b**) $S_{12}$, (**c**) $S_{21}$, and (**d**) $S_{22}$ with the frequency range spanning from 200 MHz to 65 GHz for the DUT at $V_{DS}$ = 30 V, $V_{GS}$ = −3.1 V, and $T_a$ = 200 °C.

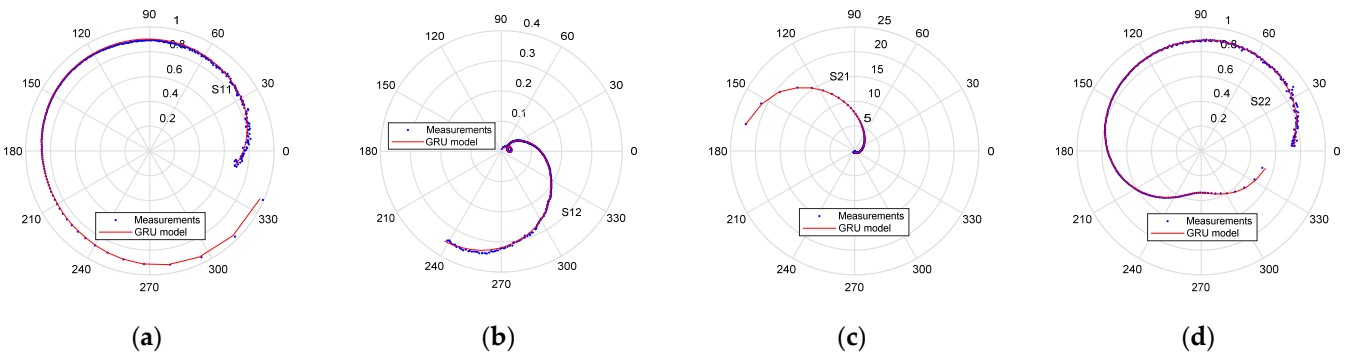

**Figure 14.** Measured (blue symbols) and GRU simulated (red lines) behavior of (**a**) $S_{11}$, (**b**) $S_{12}$, (**c**) $S_{21}$, and (**d**) $S_{22}$ with the frequency range spanning from 200 MHz to 65 GHz for the DUT at $V_{DS}$ = 30 V, $V_{GS}$ = −3.1 V, and $T_a$ = 35 °C.

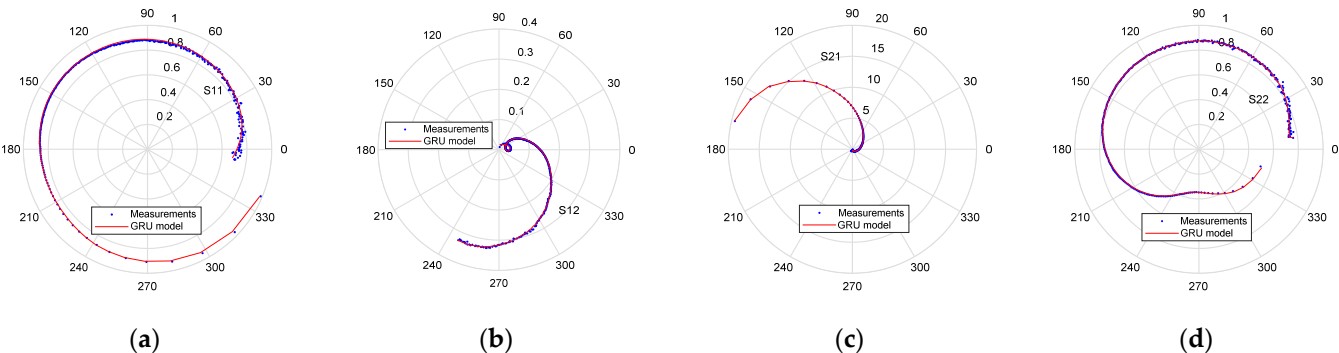

**Figure 15.** Measured (blue symbols) and GRU simulated (red lines) behavior of (**a**) $S_{11}$, (**b**) $S_{12}$, (**c**) $S_{21}$, and (**d**) $S_{22}$ with the frequency range spanning from 200 MHz to 65 GHz for the DUT at $V_{DS}$ = 30 V, $V_{GS}$ = −3.1 V, and $T_a$ = 90 °C.

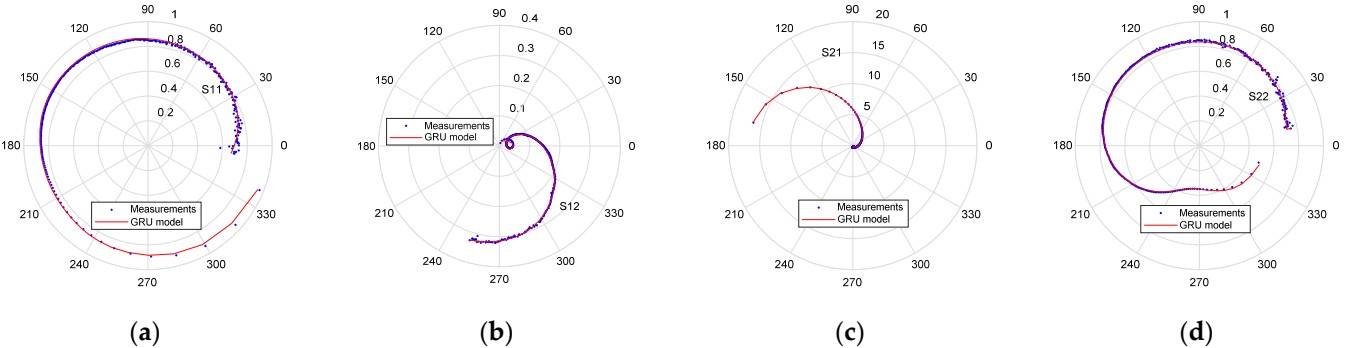

**Figure 16.** Measured (blue symbols) and GRU simulated (red lines) behavior of (**a**) $S_{11}$, (**b**) $S_{12}$, (**c**) $S_{21}$, and (**d**) $S_{22}$ with the frequency range spanning from 200 MHz to 65 GHz for the DUT at $V_{DS}$ = 30 V, $V_{GS}$ = −3.1 V, and $T_a$ = 145 °C.

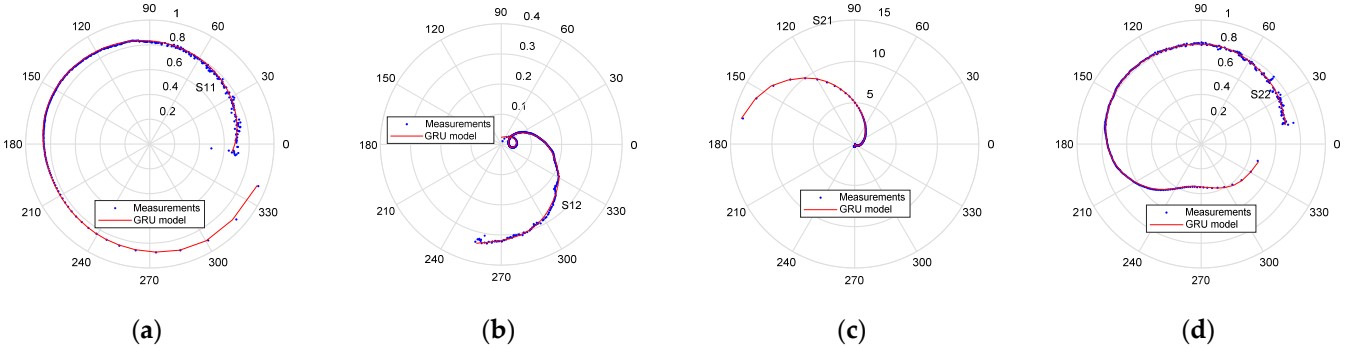

**Figure 17.** Measured (blue symbols) and GRU simulated (red lines) behavior of (**a**) $S_{11}$, (**b**) $S_{12}$, (**c**) $S_{21}$, and (**d**) $S_{22}$ with the frequency range spanning from 200 MHz to 65 GHz for the DUT at $V_{DS}$ = 30 V, $V_{GS}$ = −3.1 V, and $T_a$ = 200 °C.

As illustrated in Figures 6–9, a quite good agreement between measurements and simulations is achieved by using the equivalent-circuit model. As can be observed from Figures 10–13, a better agreement is obtained by using the ANN-based model, which has good learning ability and generalization capability, enabling the achievement of accurate predictions even for data different from those used in the training phase (i.e., at the ambient temperature of 145 °C). Figures 14–17 show that a very good agreement is also obtained by using the GRU-based model, which allows for achieving a faithful reproduction of the measured *S*-parameters.

The prediction accuracy of the extracted models is quantified by evaluating the percentage errors ($E_{ij}$) between measurements and simulations. The values of $E_{ij}$ are determined using the following formula with a total number of frequency points ($N_f$) set to 325:

$$E_{ij} = \frac{1}{N_f} \sum_f 100 \left| \frac{S_{ijMEASURED}(f) - S_{ijSIMULATED}(f)}{S_{ijMEASURED}(f)} \right| \tag{1}$$

The total percentage error $E_{TOT}$ is evaluated by averaging the four achieved values of $E_{ij}$ between measurements and simulations of $S_{11}$, $S_{21}$, $S_{12}$, and $S_{22}$:

$$E_{TOT} = \frac{(E_{11} + E_{21} + E_{12} + E_{22})}{4} \tag{2}$$

Table 3 reports the achieved values of the four $E_{11}$, $E_{12}$, $E_{21}$, $E_{22}$, and $E_{TOT}$ for all three models at the four different temperature conditions. It was found that the two behavioral models based on using ANNs and GRUs allow reaching much lower percentage errors with respect to the equivalent-circuit model. In addition, it should be underlined that the ANNs enable achieving percentage errors similar to the ones of the GRUs even for data different from those used in training (i.e., 145 °C). This clearly proves the good generalization capability of the developed ANN-based modeling approach.

**Table 3.** Percentage errors between measured and modeled scattering parameters with the frequency range spanning from 200 MHz to 65 GHz for the DUT at $V_{DS}$ = 30 V and $V_{GS}$ = −3.1 V, with four different values of $T_a$. Three models are considered: Equivalent circuit, ANNs, and GRUs.

| $T_a$ (°C) | Parameter | Equivalent Circuit | ANNs | GRUs |
|---|---|---|---|---|
| | $E_{11}$ | 45.96% | 1.04% | 1.32% |
| | $E_{21}$ | 26.86% | 12.93% | 14.63% |
| 35 | $E_{12}$ | 38.51% | 1.96% | 3.47% |
| | $E_{22}$ | 44.32% | 0.79% | 0.98% |
| | $E_{TOT}$ | 38.91% | 4.18% | 5.10% |
| | $E_{11}$ | 43.31% | 1.14% | 1.44% |
| | $E_{21}$ | 28.65% | 12.71% | 10.73% |
| 90 | $E_{12}$ | 41.38% | 2.04% | 2.69% |
| | $E_{22}$ | 45.29% | 0.78% | 0.96% |
| | $E_{TOT}$ | 39.65% | 4.17% | 3.96% |
| | $E_{11}$ | 38.62% | 2.12% | 2.10% |
| | $E_{21}$ | 27.30% | 16.51% | 19.19% |
| 145 | $E_{12}$ | 40.18% | 2.69% | 3.36% |
| | $E_{22}$ | 41.69% | 1.44% | 1.68% |
| | $E_{TOT}$ | 36.95% | 5.69% | 6.58% |
| | $E_{11}$ | 35.96% | 1.64% | 1.69% |
| | $E_{21}$ | 24.44% | 21.27% | 9.26% |
| 200 | $E_{12}$ | 38.10% | 1.91% | 2.93% |
| | $E_{22}$ | 37.88% | 1.37% | 1.34% |
| | $E_{TOT}$ | 34.10% | 6.55% | 3.81% |

Although the equivalent-circuit model has the smallest number of parameters to be optimized, it should be noted that temperature dependence is not included in the model. Therefore, the extracted equivalent-circuit elements are valid for only one temperature at

the considered bias condition. For any new temperature, the equivalent-circuit elements are to be extracted from new measurements of the *S*-parameters at the desired temperature. Therefore, the number of parameters to be optimized and the need for new measurements, and the corresponding effort for model extraction increase by increasing the number of temperatures we want to develop the model for. In contrast, the behavioral models (i.e., ANN-based and GRU-based models) have temperature dependence included. This means that once the models are developed, to obtain the *S*-parameters for any temperature, one should simply find the model response without the need for the measured *S*-parameters. Therefore, the optimization of the behavioral model parameters is performed only in the model development phase and at once for the whole temperature range with a limited number of measurements. However, in the case of the equivalent-circuit model, the extraction is to be repeated for every temperature of interest, requiring the corresponding *S*-parameter measurements.

Finally, it is worth noting that, although better usefulness comes at the cost of a lower prediction capability, the equivalent-circuit model can be seen as physically meaningful feedback for improving transistor fabrication and as the basic brick before moving on to the nonlinear and noise modeling, which is a crucial task for the design of HPAs and LNAs. To illustrate the physical meaningfulness of the obtained model parameters, Table 4 reports the impact of $T_a$ on the transconductance ($g_m$), the total gate capacitance ($C_{gg}$), and the intrinsic unity current-gain cut-off frequency ($f_T$), which is determined as the ratio between $g_m$ and $2\pi C_{gg}$. As can be observed, when heating the DUT, the degradation of the electron transport properties causes a substantial reduction in $g_m$ and, in turn, of $f_T$, which are crucial figures of merit for the assessment of the transistor performance. The obtained degradation of $g_m$ with increasing $T_a$ can be foreseen in the experiments by the observed reduction in the magnitude of the low-frequency forward transmission coefficient ($S_{21}$) when heating the DUT (i.e., the magnitude of the measured $S_{21}$ at 200 MHz decreases from 22.57 to 13.84 by increasing $T_a$ from 35 °C to 200 °C).

**Table 4.** Values of $g_m$, $C_{gg}$, and $f_T$ as a function of $T_a$ for a GaN HEMT at $V_{DS} = 30$ V and $V_{GS} = -3.1$ V.

| $T_a$ (°C) | $g_m$ (mS) | $C_{gg}$ (pF) | $f_T$ (GHz) |
|:---:|:---:|:---:|:---:|
| 35 | 347.9 | 2.560 | 21.63 |
| 90 | 293.6 | 2.372 | 19.70 |
| 145 | 249.7 | 2.222 | 17.89 |
| 200 | 215.4 | 2.130 | 16.09 |

## 4. Conclusions

A comparative study between equivalent-circuit and behavioral modeling approaches for reproducing transistor scattering parameter measurements was presented. The comparison was accomplished by using *S*-parameters measured up to 65 GHz on a $0.25 \times 1500$-μm² GaN HEMT on a silicon carbide substrate. The tested device was heated up to 200 °C and biased at $V_{DS} = 30$ V and $V_{GS} = -3.1$ V. Three different modeling strategies were analyzed: an equivalent-circuit model, artificial neural networks, and gate recurrent units. It was found that each modeling strategy possesses its own inherent strengths and weaknesses, and the best choice depends on the actual use of the model. To be sure that the selected model is the best choice, it is necessary to carefully examine the model performance in terms of prediction accuracy, generalization ability, computational efficiency, representation compactness, and model usefulness. The developed analysis showed that the equivalent circuit is a compact representation based on using sixteen parameters, which allowed for achieving a good prediction accuracy and may be very useful as physically meaningful feedback for improving device fabrication and as a starting point for building both large-signal and noise models. On the contrary, behavioral-based techniques have the great advantage of allowing achieving straightforward better prediction accuracy without needing to determine an equivalent-circuit representation. In addition, ANNs allow for ob-

taining a very good prediction accuracy even for data different from those used in training, thereby demonstrating the model generalization ability of the behavioral-based approach.

**Author Contributions:** Investigation, G.C., M.L., G.G., Z.M., J.C. and G.B.; Methodology, G.C., A.R. and N.D.; Supervision A.R., E.F. and N.D; Writing—original draft, G.C., M.L., G.G., Z.M. and G.B.; Writing—review and editing, J.C., A.R., E.F. and N.D. All authors have read and agreed to the published version of the manuscript.

**Funding:** This research was funded in part by the Italian Ministry of University and Research (MUR) through the PRIN 2017 Project under Grant 2017FL8C9N, by the GaN4AP (GaN for Advanced Power Applications) project (CUP: B82C21001520008), and by the Ministry of Science, Technological Development and Innovations of the Republic of Serbia.

**Institutional Review Board Statement:** Not applicable.

**Informed Consent Statement:** Not applicable.

**Data Availability Statement:** Not applicable.

**Conflicts of Interest:** The authors declare no conflict of interest.

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
