# Peer review of "A Comprehensive Overview of the Temperature-Dependent Modeling of the High-Power GaN HEMT Technology Using mm-Wave Scattering Parameter Measurements"

_electronics, doi:10.3390/electronics12081771_

Round 1

Reviewer 1 Report

This manuscript presents a comparative study between equivalent-circuit and behavioral modelling approaches for reproducing transistor scattering parameter measurements. Please find the comments below. Once corrected, the manuscript should be ready for publication.

1) The reviewer recommends adding references for the outstanding properties of the GaN-based materials in Line 47-52: “The great potentialities of these devices for a wide gamut of applications are due to the outstanding physical properties of the GaN-based materials, such as wide bandgap, high breakdown field, good thermal conductivity, high operation temperature, high electron mobility, high electron saturation velocity, strong spontaneous and piezoelectric polarization, and high density two-dimensional electron gas (2DEG) at the AlGaN/GaN heterointerface.”

2) There are a few typos:

Line 116: “length of 150 μm, resulting in a total gate periphery of 1.5 mm The fabrication process is”

Line 198: “Figure 4. Illustration of the internal architecture of a GRU unit.” It should be Figure 5, not Figure 4.

Reviewer 2 Report

electronics-2308830

The paper presented about a comparative study between equivalent circuit and behavioral modeling methods for reproducing transistor scattering parameter measurements. This is a valuable paper that can be officially published with minor revisions. My opinions are as follows.

1. The abstract of the paper has too much content. Can the abstract content of the paper be reduced?

2. Line121-122. The scattering parameter measurements were performed on the  DUT from 200 MHz to 65 GHz with a step of 200 MHz at four different values of Ta: 35 °C, 90 °C, 145°C, and 200 °C.  Why are these measurement conditions? Is it from previous literature? How many times are measured for each sample? Can the reliability or measurement error be provided?

Reviewer 3 Report

The manuscript represents a valuable overview of different methods for the modelling of temperature dependence of the high power GaN HEMT performance. The application of three different models is discussed including: equivalent-circuit model, artificial neural networks, and gated recurrent units. The models are compared in terms of generalizability, complexity and usefulness. The manuscript is clearly written and can be published in the present form.

Reviewer 4 Report

The manuscript is a valuable contribution to the field of s-parameter based modelling of high mobility IIIV semiconductor GaN high electron mobility transistor. The paper discusses various models and compares them. Based on the discussions, the authors indicate the suitability of different models for different applications. 

This is a minor comment to enhance the clarity of the manuscript: 

1. From figures 6-17, the authors can also include the model in the figure description. It is noted that the authors do provide this information in the main text. Given the substantial number of consecutive results, this will greatly improve the readability of the manuscript. 

2. 
